# Home delivery among women who had optimal ANC follow-up in Sub-Saharan Africa: A multilevel analysis

**Alebachew Ferede Zegeye**[1]*, **Wubshet Debebe Negash**[2], **Alemneh Tadesse Kassie**[3], **Likinaw Abebaw Wassie**[1], **Tadesse Tarik Tamir**[4]

**1** Department of Medical Nursing, School of Nursing, College of Medicine and Health Sciences, University of Gondar, Gondar, Ethiopia, **2** Department of Health Systems and Policy, Institute of Public Health, College of Medicine and Health Sciences, University of Gondar, Gondar, Ethiopia, **3** Department of Clinical Midwifery, School of Midwifery, College of Medicine and Health Sciences, University of Gondar, Gondar, Ethiopia, **4** Department of Pediatric and Child Health Nursing, School of Nursing, College of Medicine and Health Sciences, University of Gondar, Gondar Ethiopia

* alexferede24@gmail.com

**Data Availability Statement:** The most recent data from the Demographic and Health Survey were used in this study, and it is publicly available online at (https://www.dhsprogram.com).

## Abstract

### Background

Home deliveries, where most births take place, are significantly responsible for the majority of maternal mortality. In order to develop appropriate policies and methods that could aid in addressing the issue, it is important to understand the scope of home delivery and its determinants in developing countries. Therefore, this study aims to ascertain the prevalence and factors associated with home delivery among women who had optimal ANC follow up in the Sub-Saharan Africa countries.

### Methods

A population based cross-sectional study was conducted. Data from the most recent Demographic and Health Surveys, which covered 23 Sub-Saharan Africa countries from 2014 to 2020, were used. The study used a total of 180,551 women who had optimal ANC follow up weighted sample. Stata 14 was used to analyze the data. The determinants of home delivery were determined using a multilevel mixed-effects logistic regression model. Factors associated with home delivery in the multilevel logistic regression model were declared significant at p-values <0.05. The adjusted odds ratio and confidence interval were used to interpret the results.

### Results

In Sub-Saharan Africa, three in ten (30%) women who had optimal ANC follow-up gave birth at home. Individual-level variables such as maternal age (20 to 35 years) (AOR = 1.27, 95% CI: 1.10, 1.46), no formal education (AOR = 3.10, 95% CI: 2.68, 3.59), pregnancy complications (AOR = 0.74, 95% CI: 0.67, 0.82), distance to a health facility (AOR = 1.43, 95% CI: 1.30, 1.58), and poor wealth status (AOR = 2.71, 95% CI: 2.37, 3.10) had higher odds of home delivery. Community-level variables such as rural residence (AOR = 2.83, 95% CI:

**Funding:** The authors received no specific funding for this work.

**Competing interests:** The authors declared that there is no competing interest.

2.48, 3.22), living in central Sub-Saharan Africa (AOR = 7.95, 95% CI: 5.81, 10.9), and eastern Sub-Saharan Africa (AOR = 2.74, 95% CI: 2.09, 3.59), were significantly associated with home delivery.

## Conclusions and recommendation

This study concludes that home delivery in sub-Saharan Africa among women who had optimal ANC follow-up were high. The study identified that both individual and community-level variables were determinants of home delivery. Therefore, the Government and ministries of health in Sub-Saharan Africa countries should give attention to those women who reported distance as a big problem to health facilities and for rural resident women while designing policies and strategies targeting reducing home delivery in sub-Saharan Africa.

## Background

Home delivery refers to having a baby in an unclean, non-clinical setting [1, 2]. Millions of home births occur each year without the assistance of medical professionals with the necessary training. According to the studies, just 16% of deliveries were attended by qualified professionals, while the vast majority (78%) was attended by traditional birth attendants. In developed countries, more than 99% of deliveries are assisted by skilled health professionals, compared to 62% in developing countries [3].

Globally, maternal mortality continues to be a serious issue for public health. The greatest burden resides in sub-Saharan African (SSA) regions, where 85% of maternal deaths have been reported. According to studies in the first week of life, 4 million newborns and 529,000 mothers worldwide die [4, 5]. Obstructed Labor, uterine rupture, severe pre- and eclampsia, malaria, and abortion-related complications are the leading causes of maternal death in countries with low incomes [6, 7]. Home deliveries, where most births take place, are significantly responsible for the majority of maternal mortality in sub-Saharan Africa. Many deliveries still take place at home in low- and middle-income (LMIC) countries without the aid of skilled health care workers [5].

Home deliveries have an adverse effect on the child and are responsible for newborn morbidity and mortality. Deliveries at home are attended by unskilled healthcare workers and take place in a potentially hazardous setting, which increases the risk of postpartum haemorrhage and infection to family members or traditional birth attendants who assist births without protective equipment [8]. Studies showed that home delivery has been associated with young maternal age, low educational attainment, residing in a rural area, low socioeconomic status, timing of first ANC services, distance to health care, and pregnancy complications [9, 10].

Despite the fact that professional delivery attendance can save women's lives, only 59% of births in sub-Saharan Africa between 2012 and 2017 were attended by experienced birth attendants. A potential cause to the high maternal mortality rate in this region is the substantial number of home deliveries. The primary way to reduce the considerable number of direct causes of maternal death, which include obstetric complications like haemorrhage, pregnancy-induced hypertension, sepsis, and obstructed labor that account for 64% of maternal deaths, is to have the delivery attended by a skilled birth attendant in the health care facility [11, 12].

The overall scope of home delivery and its determinants are still unknown, despite the significant proportion of maternal morbidity and mortality in sub-Saharan African countries that

is overwhelmingly attributable to home delivery. Women who attended ANC would be more aware of the advantages of facility delivery and would make early arrangements to give birth in a healthcare facility.

However, the prevalence and contributing factors of home birth among women who have optimal ANC follow-up in sub-Saharan African countries are not well studied. So this study aimed to assess the prevalence of home delivery in sub-Saharan Africa and its associated factors among women with optimal ANC follow-up by using the recent Demographic and Health Survey (DHS) data through conceptual stratification of variables. Furthermore, the current study's findings offer support for health planners, policymakers, sponsors, and health professionals as they desire to further reduce home deliveries, which would assist in lowering maternal mortality in countries with middle and low incomes, such as sub-Saharan Africa.

## Methods

### Study design, study area, and period

A population based cross-sectional study was conducted. A recent DHS survey Data from the 23 sub-Saharan African countries, which were conducted between 2014 and 2020, were used to carry out multilevel mixed effect analysis. To produce updated health and health-related indicators, the DHS which is a community-based survey is carried out every five years.

### Data source, study population and sampling technique

The most recent sub-Saharan African countries Demographic Health Survey (DHS) datasets from 2014 to 2020 were used for the secondary data analysis. We used DHS surveys from 23 sab-Saharan Africa countries such as Angola, Benin, Burundi, Cameron, Ethiopia, Ghana, Gambia, Guinea, Kenya, Liberia, Lesotho, Mali, Malawi, Nigeria, Rwanda, Serra Leone, Senegal, Chad, Tanzania, Uganda, South Africa, Zambia, and Zimbabwe. The data were appended to figure out the prevalence and contributing factors of home delivery among women receiving the optimal ANC follow-up in sub-Saharan African nations. The survey for every country contains different datasets, including those for males, women, children, births, and households. DHS deploys a stratified two-stage cluster design that includes enumeration areas as the first stage and generates a sample of households from each enumeration area as the second stage. The variable place of delivery (m15) from the maternal record (IR) data set was recoded to create the outcome variable (home delivery). A binary logistic regression model was used to determine the factors associated to home delivery. Determinants of home delivery were reported in terms of an adjusted odds ratio (AOR) with a significance level of (95%). In the univariate analysis, at 95% confidence intervals with a p-value of $< 0.25$ was considered a candidate for the multivariable analysis of data. All variables with p values $<0.05$ were considered statistically significant. A total weighted sample of 180,551 women was included in the study (**Table 1**).

### Study variables

**Dependent variables.**   In this study, the dependent variable was home delivery. It was recorded as "Yes = 1" if the delivery took place at the house of a respondent or someone else. On the other hand, it was recorded as "No = 0" if the place of delivery was a healthcare facility (such as a government hospital, health center, health post, private clinic, or private hospital) [2].

**Independent variables.**   Since DHS data are hierarchical, independent variables from two sources (variables at the individual and community levels) were taken into consideration for

**Table 1. Sample size for prevalence and determinants of home delivery among pregnant women with optimal ANC follow-up in Sub-Saharan Africa countries.**

| Country | Year of survey | Weighted sample (n) | Weighted sample (%) |
|---|---|---|---|
| Angola | 2015 | 10,270 | 5.68 |
| Benin | 2017/18 | 9,062 | 5.01 |
| Burundi | 2016/17 | 8,472 | 4.69 |
| Cameron | 2018 | 7,402 | 4.09 |
| Ethiopia | 2016 | 5,976 | 3.31 |
| Ghana | 2015 | 5,266 | 2.91 |
| Gambia | 2019 | 7,094 | 3.92 |
| Guinea | 2018 | 4,328 | 2.39 |
| Kenya | 2014 | 13,975 | 7.74 |
| Liberia | 2019/20 | 5,014 | 2.77 |
| Lesotho | 2014 | 2,456 | 1.36 |
| Mali | 2018 | 6,226 | 3.44 |
| Malawi | 2015 | 10,427 | 5.82 |
| Nigeria | 2018 | 24,058 | 13.31 |
| Rwanda | 2019/20 | 4,798 | 2.65 |
| Serra Leone | 2019 | 8,373 | 4.63 |
| Senegal | 2019 | 4,055 | 2.24 |
| Chad | 2014/15 | 10,637 | 5.88 |
| Tanzania | 2015 | 6,575 | 3.64 |
| Uganda | 2016 | 11,238 | 6.22 |
| South Africa | 2016 | 2,762 | 1.58 |
| Zambia | 2018 | 7,234 | 4 |
| Zimbabwe | 2015 | 4,853 | 2.68 |
| Total sample size | | 180,551 | 100 |

this analysis. The individual-level independent variables were Maternal age (15–19, 20–35, 36–49), Maternal education (No formal education, Primary, Secondary and Higher), Maternal occupation (House wife, Governmental employee, Others,), Religion (Orthodox, Catholic, Protestant, Muslim, Others), Marital status of the mother (Never married, Currently married, Formerly/ever married), Sex of child (Male, Female), Birth weight (Low, Normal, High), Place of delivery (Home, Facility), Mode of delivery CS (Yes, No), Pregnancy complications (No, Yes), Sex of household head (Male, Female), Distance to health facility (Big problem, Not a big problem), Timing of first antenatal checkup (First trimester, Second trimester, Third trimester), Total children ever born (1–3, 4–6, 7–9, >9), and household wealth index (Poor, Middle, Rich).

The community-level variables were Place of residence (Urban, Rural), Community illiteracy (Low, High), Community-level poverty (Low, High), Community media exposure (Low, High) Country category (Central, Eastern, West, Southern Sub-Saharan).

Data leveling of the dependent and independent variables (both individual and community-level variables) included in this study was made based on conceptual stratification using known facts and the existing literatures, which have similar objectives with the current study.

## Data processing and statistical analysis

The data that were obtained from recent DHS data sets were cleaned, recorded, and analyzed using STATA version 14 statistical software. The DHS data's variables are organized in

clusters, and those in a cluster are more similar to one another than those of other clusters. To employ a standard logistic regression model, the assumptions of independent observations and equal variance across clusters were broken. This suggests that using a sophisticated model to take into account between-cluster factors is necessary. Given this, multilevel mixed-effects logistic regression was used to determine the factors that associated with home delivery. Multilevel mixed effect logistic regression follows four models: the null model (outcome variable only), mode I (only individual level variables), model II (only community level variables), and model III (both individual and community level variables). The model without independent variables (null model) was used to check the variability of home delivery across the cluster. The association of individual-level variables with the outcome variable (Model I) and the association of community-level variables with the outcome variable (Model II) were assessed. In the final model (Model III), the association of both individual and community-level variables was fitted simultaneously with the outcome variable (home delivery).

### Ethical approval and consent to participate

After the consent manuscript was submitted to the DHS Programme/ICF to download the dataset for this investigation, the International Review Board of Demographic and Health Surveys (DHS) programme data archivists' waived informed consent. Since the study's data came from a publicly accessible source, it is not an experimental study. All the methods were conducted according to the Helsinki Declarations. More details regarding DHS data and ethical standards are available online at (http://www.dhsprogram. com).

### Random effects

Random effects or measures of variation of the outcome variables were estimated by the median odds ratio (MOR), intra-class correlation coefficient (ICC), and proportional change in variance (PCV). The intra-class correlation coefficient (ICC) and proportional change in variance (PCV) were computed to measure the variation between clusters. Taking clusters as a random variable, the ICC reveals the variation of home delivery between clusters is computed as; $ICC = \frac{VC}{VC+3.29} \times 100\%$. The MOR is the median value of the odds ratio between the area of the highest risk and the area of the lowest risk for home delivery when two clusters are randomly selected, using clusters as a random variable; $MOR = e^{0.95\sqrt{VC}}$.

Moreover, the PCV demonstrates the variation in the prevalence of home delivery explained by factors and computed as; $PCV = \frac{Vnull-VC}{Vnull} \times 00\%$; where Vnull = variance of the null model and VC = cluster level variance [13–15]. The fixed effects were used to estimate the association between the likelihood of home delivery and individual and community level independent variables. It was assessed and the strength was presented using adjusted odds ratio (AOR) and 95% confidence intervals with a p-value of < 0.05. Because of the nested nature of the model, deviation = -2(log likelihood ratio) was used to compare models, and the model with the lowest deviance was selected as the best-fit model. The variables used in the models were verified for multi-collinearity by measuring the variance inflation factors (VIF), with the findings falling within acceptable limits of 1–10.

## Result

### Sociodemographic and economic characteristics of women who had optimal ANC follow up in Sub-Saharan Africa countries

A total of 180,551 women who had optimal ANC follow-up were included in this study. Greater than one-third of women (35.52%) did not have formal education. Nearly one-third

(33.30%) of the participants were living in rural areas of sub-Saharan Africa countries, and 62,208 (37.66%) were living in sub-Saharan Africa countries where distance to health facilities is a big problem to get antenatal, prenatal, and postnatal services. About more than half (52.69%) of women living in sub-Saharan African countries have poor community media exposure (**Table 2**).

## Prevalence of home delivery among women who had optimal ANC follow-up in Sub-Saharan Africa countries

The prevalence of home delivery among women who had optimal ANC follow-up in sub-Saharan African countries was 31.18% (95% CI: (30.9, 31.40)). The magnitude of urban and rural home delivery in sub-Saharan African countries was found to be 18.5% and 81.5%, respectively (**Fig 1**). West sub-Saharan Africa (40.3%) and Southern sub-Saharan Africa (1.1%) countries had the highest and lowest rates of home delivery, respectively (**Fig 2**).

## Random effect ((Measures of variation) and model fitness

A null model was used to determine whether the data supported the decision to assess randomness at the community level. Findings from the null model showed that there were significant differences in home delivery between communities, with a variance of 0.5679354 and a P value of 0.000. The variance within clusters contributed 85.28% of the variation in home delivery, while the variance across clusters was responsible for 14.72% of the variation. In the null model, the odds of a home delivery differed between higher and lower risk clusters by a factor of 2.05 times. The intraclass correlation value for Model I indicated that 13.14% of the variation in home delivery accounts for the disparities between communities. Then, with the null model, we used community-level variables to generate Model II. Cluster variations were the basis for 14.20% of the differences in home delivery, according to the ICC value from Model II. In the final model (model III), which attributed approximately 33.05% of the variation in the likelihood of home delivery to both individual and community-level variables (**Table 3**), the likelihood of home delivery varied by 1.80 times across low and high home delivery clusters.

## Association of individual and community level factors with home delivery among women who had optimal ANC follow up in the sub-Saharan Africa countries

In the final fitted model of multivariable multilevel logistic regression, maternal age 20–35, maternal education, presence of pregnancy complications, distance to a health facility, wealth status, place of residence, and country category (central and eastern Africa) were significantly associated with home delivery among women with optimal ANC follow-up (**Table 4**).

The odds of home delivery were 1.27 times higher among pregnant women aged 20 to 35 years compared to women aged 36 to 49 years (AOR = 1.27, 95% CI: 1.10, 1.46). Home delivery was 3.10 and 2.20 times more likely to occur among women who did not have formal education and primary education than among women who have secondary and above-level education (AOR = 3.10, 95% CI: 2.68, 3.59) and 2.20, 95% CI: 1.92, 2.51), respectively. A woman whose occupation is housewife was 1.16 times more likely to deliver at home compared to a woman whose occupation is government employee (AOR = 1.16, 95% CI: 1.04, 1.28).

The odds of home delivery were 26% less likely to occur among women having pregnancy complications compared to women not having pregnancy complications (AOR = 0.74, 95% CI: 0.67, 0.82). The odds of home delivery were 1.43 times more likely to occur among women

**Table 2. Sociodemographic and economic characteristics of women who had optimal ANC follow up in Sub-Saharan Africa countries.**

| Variables | Frequency (n) | Percent (%) |
|---|---|---|
| **Individual level variables** | | |
| Maternal age | | |
| 15–19 | 9,089 | 5.03 |
| 20–35 | 139,058 | 76.93 |
| 36–49 | 32,615 | 18.04 |
| Maternal educational level | | |
| No formal education | 64,208 | 35.52 |
| primary | 61,527 | 34.04 |
| Secondary and higher | 55,027 | 30.44 |
| Maternal occupation | | |
| House wife | 46,656 | 28.16 |
| Governmental employee | 114,328 | 69.01 |
| Others | 4,673 | 2.82 |
| Religion | | |
| Orthodox | 57,581 | 31.85 |
| Catholic | 46,861 | 25.92 |
| Protestant | 31,387 | 17.36 |
| Muslim | 10,125 | 5.6 |
| Others | 34,808 | 19.26 |
| Marital status of the mother | | |
| Never married | 10,516 | 5.82 |
| Currently married | 158,650 | 87.77 |
| Formerly/ever married | 11,596 | 6.42 |
| Pregnancy complications | | |
| No | 5,884 | 33.85 |
| Yes | 11,402 | 65.59 |
| Sex of household head | | |
| Male | 142,754 | 78.97 |
| Female | 38,008 | 21.03 |
| Distance to health facility | | |
| Big problem | 62,208 | 37.66 |
| Not a big problem | 102,995 | 62.34 |
| Timing of first antenatal check up | | |
| First trimester | 51,294 | 51.44 |
| Second trimester | 46,892 | 47.02 |
| Third trimester | 1,539 | 1.54 |
| Total children ever born | | |
| 1–3 | 92,919 | 51.40 |
| 4–6 | 60,926 | 33.71 |
| 7–9 | 22,074 | 12.21 |
| >9 | 4,843 | 2.68 |
| Wealth index | | |
| Poor | 80,873 | 44.74 |
| Middle | 36,158 | 20.00 |
| Rich | 63,731 | 35.26 |
| **Community level variables** | | |

(*Continued*)

**Table 2.** (Continued)

| Variables | Frequency (n) | Percent (%) |
|---|---|---|
| Place of residence | | |
| Rural | 60,200 | 33.30 |
| Urban | 120,562 | 66.70 |
| Community media exposure | | |
| Low | 95,235 | 52.69 |
| High | 85,527 | 47.31 |
| Community poverty | | |
| Low | 90,883 | 50.28 |
| High | 89,879 | 49.72 |
| Community illiteracy | | |
| Low | 80,695 | 44.64 |
| High | 100,067 | 55.36 |
| Country category | | |
| Central | 28,309 | 15.66 |
| Eastern | 73,659 | 40.75 |
| West | 73,476 | 40.65 |
| Southern | 5,318 | 2.94 |

whose distance to a health facility is a big problem compared to women whose distance to a health facility is not a big problem (AOR = 1.43, 95% CI: 1.30, 1.58).

Home delivery was 2.71 times more likely to occur among women living in a poor wealth status compared to women living in a rich wealth index (AOR = 2.71, 95% CI: 2.37, 3.10). The odds of home delivery were 2.83 times higher to occur among women in rural residences compared to women living in urban areas (AOR = 2.83, 95% CI: 2.48, 3.22). The odds of home delivery were 7.95 and 2.74 times more likely to occur among women living in central and eastern Africa compared to women living in southern Africa (AOR = 7.95, 95% CI: 5.81, 10.9) and (AOR = 2.74, 95% CI: 2.09, 3.59), respectively.

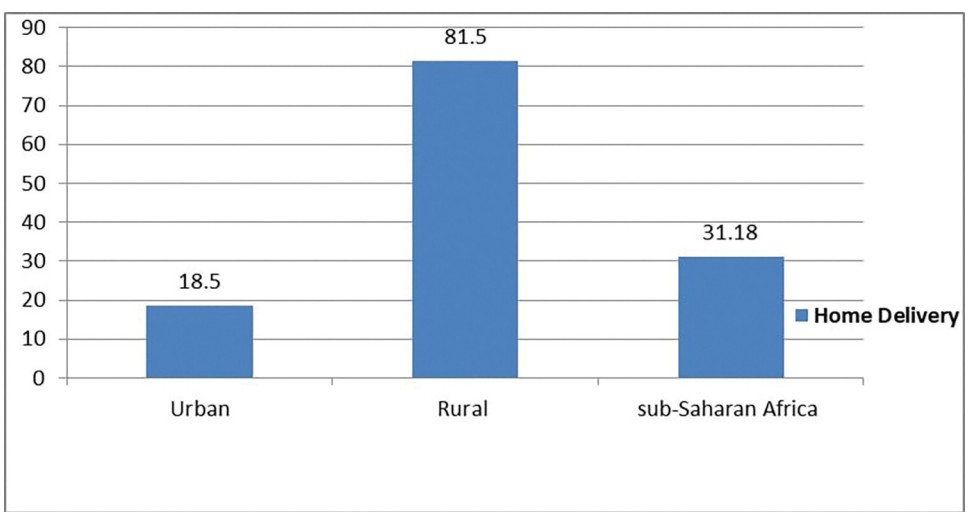

**Fig 1. Prevalence of home delivery among women who had optimal ANC follow-up in Sub-Saharan Africa countries.**

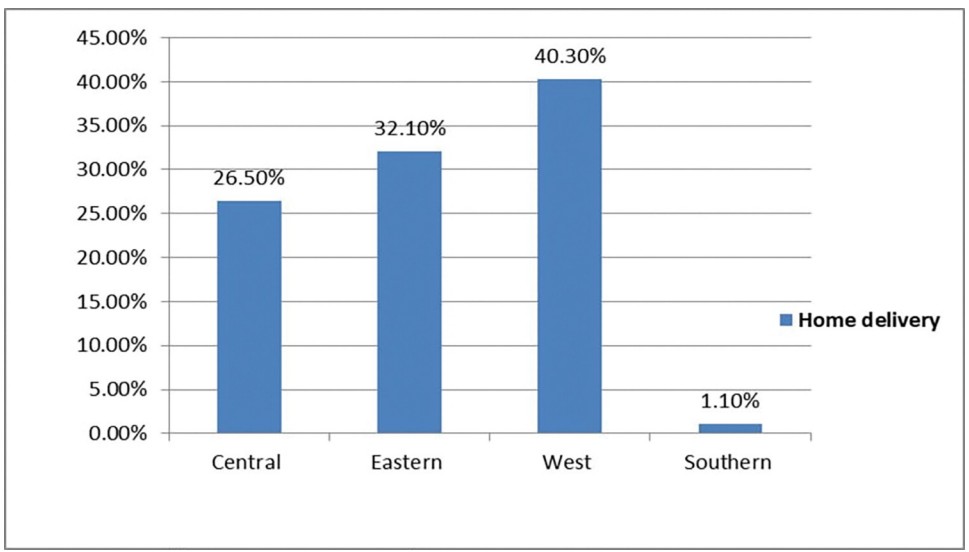

**Fig 2. Regional prevalence of home delivery among women who had optimal ANC follow-up in Sub-Saharan Africa countries.**

## Discussion

Home deliveries significantly contribute to a surge in both maternal and child mortality in developing countries such as Sub-Saharan Africa. The purpose of this study was to determine the prevalence and contributing factors of home delivery among pregnant women in sub-Saharan African countries who experienced optimal ANC follow-up.

In this study, the prevalence of home delivery among women who experienced optimal ANC follow-up was found to be 31.18% (95% CI: (30.9, 31.40). The finding is higher than the previous studies conducted in India which is 15.4% [16], Philippines 23.53% [17]. The higher prevalence of home delivery in this study than previous findings in India and the Philippines could be due to differences in socio-economic status and variability in health infrastructure and health system policy, women's attitudes towards place of delivery, and cultural differences across countries. The health service coverage, quality of maternal healthcare services, and economic and health policies of India and the Philippines are better compared with those of Sub-Saharan Africa countries and might play a role in reducing the likelihood of home delivery.

On the other hand, the prevalence of home delivery in this study was lower than the findings conducted in Ethiopia, which were 35.2% and 80% [18, 19], Liberia, 90.6% [20], Tanzania,

**Table 3. Model comparison and random effect analysis for home delivery among women who had optimal ANC follow-up in Sub-Saharan Africa countries.**

| Parameter | Null model | Model I | Model II | Model III |
|---|---|---|---|---|
| Variance | 0.5679354 | 0.4978602 | 0.5443848 | 0.3802187 |
| ICC | 14.72% | 13.14% | 14.20% | 10.34% |
| MOR | 2.05 | 1.96 | 2.02 | 1.80 |
| PCV | Reference | 12.34% | 4.15% | 33.05% |
| **Model fitness** | | | | |
| LLR | -108603.34 | -6595.0434 | -98014.832 | -6220.2068 |
| Deviance | 217,206.68 | 13,190.0868 | 196029.664 | 12,440.4136 |

ICC: interacluster correlation, LLR: logliklihood ratio, MOR: median odds ratio, PCV: proportional change in variance.

**Table 4. Multivariable multilevel logistic regression analysis of individual-level and community level factors associated with home delivery among women who had optimal ANC follow up in sub Saharan Africa, DHS 2014–2020.**

| Individual and community level factors | Model I AOR(95% CI) | Model II AOR(95% CI) | Model III AOR(95% CI) |
|---|---|---|---|
| Maternal age | | | |
| 15–19 | 1.29 (1.01, 1.64) | | 0.88 (0.68, 1.13) |
| 20–35 | 1.48 (1.29, 1.69) | | **1.27 (1.10, 1.46)** |
| 36–49 | 1 | | 1 |
| Maternal educational level | | | |
| No formal education | 3.57 (3.10, 4.11) | | **3.10 (2.68, 3.59)** |
| primary | 2.60 (2.23, 2.90) | | **2.20 (1.92, 2.51)** |
| Secondary and above | 1 | | 1 |
| Maternal occupation | | | |
| House wife | 0.15 (1.05, 1.27) | | 0.16 (1.04, 1.28) |
| Governmental employee | 1 | | 1 |
| Others | 1.08 (0.65, 1.80) | | 1.06 (0.63, 1.79) |
| Religion | | | |
| Orthodox | 1 | | 1 |
| Catholic | 1.69(0.47, 1.95) | | 1.05 (0.91, 1.23) |
| Protestant | 3.04(0.61, 3.55) | | 1.63 (0.38, 1.94) |
| Muslim | 2.02 (0.69, 2.40) | | 1.46 (0.21, 1.76) |
| Others | 1.53 (0.34, 1.75) | | 2.66 (0.28, 3.10) |
| Marital status of the mother | | | |
| Never married | 0.92 (0.69, 1.21) | | 1.03 (0.78, 1.37) |
| Currently married | 1 | | 1 |
| Formerly/ever married | 0.92 (0.77, 1.10 | | 0.94 (0.78, 1.13) |
| Pregnancy complications | | | |
| No | 1 | | 1 |
| Yes | 0.66 (0.60, 0.73) | | **0.74 (0.67, 0.82)** |
| Sex of household head | | | |
| Male | 1.05 (0.94, 1.17) | | 0.96 (0.86, 1.08) |
| Female | 1 | | 1 |
| Distance to health facility | | | |
| Big problem | 1.80 (1.63, 1.98) | | **1.43 (1.30, 1.58)** |
| Not a big problem | 1 | | 1 |
| Timing of first antenatal checkup | | | |
| First trimester | 1 | | 1 |
| Second trimester | 0.43 (0.30, 1.57) | | 0.26 (0.14, 1.39) |
| Third trimester | 1.06 (0.69, 1.62) | | 1.13 (0.74, 1.73) |
| Total children ever born | | | |
| 1–3 | 1 | | 1 |
| 4–6 | 0.48 (0.33, 1.65) | | 0.35 (0.21, 1.52) |
| 7–9 | 1.23 (1.88, 2.65) | | 1.69 (0.41, 2.02) |
| >9 | 0.89 (0.09, 4.01) | | 0.85 (0.32, 2.58) |
| Wealth index | | | |
| Poor | 3.21(2.85, 3.61) | | **2.71 (2.37, 3.10)** |
| Middle | 0.96(0.71, 2.25) | | 0.69 (0.45, 1.96) |
| Rich | 1 | | 1 |
| **Community level variables** | | | |

*(Continued)*

**Table 4.** (Continued)

| Individual and community level factors | Model I AOR(95% CI) | Model II AOR(95% CI) | Model III AOR(95% CI) |
|---|---|---|---|
| Place of residence | | | |
| Rural | | 4.22 (4.10, 4.34) | **2.83 (2.48, 3.22)** |
| Urban | | 1 | 1 |
| Community media exposure | | | |
| Low | | 1.11 (0.99, 1.23) | 1.07 (0.93, 1.22) |
| High | | 1 | 1 |
| Community poverty | | | |
| Low | | 0.60 (0.55, 1.67) | 0.84 (0.73, 1.96) |
| High | | 1 | 1 |
| Community illiteracy | | | |
| Low | | 1.01 (0.91, 1.12) | 1.01 (0.88, 1.16) |
| High | | 1 | 1 |
| Country category | | | |
| Central | | 12.3 (11.2, 13.5) | **7.95 (5.81, 10.9)** |
| Eastern | | 1.96 (1.79, 2.14) | **2.74 (2.09, 3.59)** |
| West | | 3.76 (3.44, 4.11) | 0.83 (0.64, 1.08) |
| Southern | | 1 | 1 |

35.5% [21]. The differences between prevalence in our study and that in Ethiopia, Liberia, and Tanzania may be due to the fact that our study used secondary data from the DHS report and was conducted at the level of sub-Saharan African countries, whereas the latter studies used primary data sources. Additionally, in sub-Saharan Africa, there are some countries that have good health infrastructure and quality maternal healthcare services compared to those of individual countries.

In the multivariable multilevel mixed effect logistic regression analysis, maternal age, maternal education, presence of pregnancy complications, distance to a health facility, wealth status, place of residence, and country category (central and eastern Africa) were significantly associated with home delivery among women who had optimal ANC follow-up.

In this study, the odds of home delivery were 1.27 times higher among pregnant women aged 20 to 35 years compared to women aged 36 to 49 years. This finding is in line with previous findings [5, 22]. On the one hand, it's possible that the relationship between home birth and being in the 20–35 year old maternal age range is because the woman may not have been psychologically prepared to give birth in a health care facility. On the other hand, young women who have never given birth before do not consult their parents about where to deliver the baby. The finding of this study is contradictory with the study conducted in [20]. This could be because older women who have given birth more than once think they are more experienced at giving birth, which makes them more interested in choosing home delivery services.

The odds of home delivery among women who had optimal ANC follow up were 3.10 and 2.20 times higher among women who did not have formal education and primary education than among women who have secondary and above-level education respectively. This is consistent with previous studies [1, 23, 24]. This is possibly related to the fact that educated women have information on the benefits of deliveries in health care institutions, and this could help with behavioral changes that might allow women to accept and utilise maternal health services. This implies that highly educated women pay more attention to the potential risks and challenges of home delivery and have greater knowledge about institutional delivery.

The odds of home delivery were 26% less likely to occur among women having pregnancy complications compared to women not having pregnancy complications. This finding was coherent with previous studies [25, 26]. The possible rationale for the association might be that confronting complications during pregnancy causes the women to utilise different maternal health services, such as antenatal services, which could help them with birth preparedness and complication readiness, which in turn can increase institutional delivery. Women who experienced complications can have more practical experience in life-threatening conditions than those who did not, which might encourage them to give birth under the supervision of experts who can help them during an emergency in case it happens. Moreover, coming across pregnancy complications can make women seek health care services during pregnancy, and they can have a probability of being advised on facility delivery by health experts [27, 28].

The odds of home delivery were 1.43 times more likely to occur among women whose distance to a health facility is a big problem compared to women whose distance to a health facility is not a big problem. This study finding is supported by the previous findings [29, 30]. The possible explanation might be that if the distance to a health facility is considered a barrier, women are less likely to utilise health services as a place of delivery due to a lack of transportation fees and easy transportation services to the health facility. In this study, the wealth status of women was another factor significantly associated with home delivery. Home delivery was 2.71 times more likely to occur among women living in a poor wealth status compared to women living in a rich wealth status. This finding is supported by the study findings [31–33]. This is due to the high costs associated with giving birth in a healthcare facility; women of low socioeconomic status are more likely to choose home delivery. Since institutional delivery requires costs associated with pregnancy and birth [34, 35].

The odds of home delivery were 2.83 times higher to occur among women in rural residences compared to women living in urban areas. This is consistent with the findings from previous studies [36, 37]. The possible explanation might be that those women living in rural areas of Sub-Saharan Africa countries have low financial capability to afford transportation costs, poor knowledge of institutional delivery services, and less availability of nearby health care services, which leads them to prefer home delivery.

Geographical region was significantly associated with home delivery among women who had optimal ANC follow up in Sub-Saharan Africa countries. The odds of home delivery were 7.95 and 2.74 times more likely to occur among women living in central and eastern Sub-Saharan Africa compared to women living in southern Africa Sub-Saharan Africa respectively. This might be related to the difference in the availability of health facilities. In particular, southern sub-Saharan Africa, where the quality of maternal health services is better than in central and eastern sub-Saharan Africa, Moreover, women from southern sub-Saharan Africa are also more informed about the risks of home delivery, which could reduce the likelihood of delivering at home [38].

## Strength and limitations

The study's strength was the utilisation of recently conducted large-sample national demography and health surveys from sub-Saharan African countries. The representativeness of our finding, however, may have been impacted by the fact that some sub-Saharan African countries have not carried out a demographic and health survey since 2014. Moreover psychological factors such as feeling less fear at home, more comfort of the mother at home and perhaps cultural factors involved in choosing to give birth at home were not included in our study.

## Conclusions and recommendation

This study concludes that home delivery rates in sub-Saharan Africa among women who had optimal ANC follow-up were high. The study identified that both individual and community-

level variables were determinants of home delivery. Therefore, the Government and ministry of health in Sub-Saharan Africa countries should give attention to those women who reported distance as a big problem to health facilities and for rural resident women while designing policies and strategies targeting reducing home delivery in sub-Saharan Africa.

## Acknowledgments

We are grateful to the DHS programmes for letting us use the relevant DHS data in this study.

## Author Contributions

**Conceptualization:** Alebachew Ferede Zegeye, Likinaw Abebaw Wassie, Tadesse Tarik Tamir.

**Data curation:** Alebachew Ferede Zegeye, Wubshet Debebe Negash, Tadesse Tarik Tamir.

**Formal analysis:** Alebachew Ferede Zegeye, Wubshet Debebe Negash, Alemneh Tadesse Kassie.

**Funding acquisition:** Wubshet Debebe Negash, Alemneh Tadesse Kassie, Likinaw Abebaw Wassie, Tadesse Tarik Tamir.

**Investigation:** Alebachew Ferede Zegeye.

**Methodology:** Alebachew Ferede Zegeye, Wubshet Debebe Negash.

**Project administration:** Alebachew Ferede Zegeye.

**Resources:** Wubshet Debebe Negash.

**Software:** Alebachew Ferede Zegeye, Wubshet Debebe Negash, Alemneh Tadesse Kassie, Tadesse Tarik Tamir.

**Supervision:** Alemneh Tadesse Kassie, Tadesse Tarik Tamir.

**Validation:** Alebachew Ferede Zegeye, Alemneh Tadesse Kassie, Likinaw Abebaw Wassie, Tadesse Tarik Tamir.

**Visualization:** Tadesse Tarik Tamir.

**Writing – original draft:** Alebachew Ferede Zegeye, Likinaw Abebaw Wassie, Tadesse Tarik Tamir.

**Writing – review & editing:** Alebachew Ferede Zegeye, Likinaw Abebaw Wassie, Tadesse Tarik Tamir.

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
