## [Decision Letter · Decision Letter 0]

23 Oct 2023

PONE-D-23-19452Prevalence and determinants of home delivery among women who had optimal ANC follow-up in Sub-Saharan Africa countries: A Multilevel Analysis of a Recent Demographic and Health SurveyPLOS ONE

Dear Dr. Alebachew,

Thank you for submitting your manuscript to PLOS ONE. After careful consideration, we feel that it has merit but does not fully meet PLOS ONE’s publication criteria as it currently stands. Therefore, we invite you to submit a revised version of the manuscript that addresses the points raised during the review process. Please submit your revised manuscript by **8th of November 2023**. If you will need more time than this to complete your revisions, please reply to this message or contact the journal office at plosone@plos.org. Please include the following items when submitting your revised manuscript:A **rebuttal letter that responds to each point** raised by the academic editor and reviewer(s). You should upload this letter as a separate file labeled 'Response to Reviewers'.A marked-up copy of your manuscript that highlights changes made to the original version. You should upload this as a separate file labeled '**Revised Manuscript with Track Changes**'.An unmarked version of your revised paper without tracked changes. You should upload this as a separate file labeled '**Manuscript**'.

We look forward to receiving your revised manuscript.

Kind regards,

Kahsu Gebrekidan

Academic Editor

PLOS ONE

Journal Requirements:

- https://www.frontiersin.org/articles/10.3389/fpubh.2022.862616/full

- https://pubmed.ncbi.nlm.nih.gov/36906518/

In your revision ensure you cite all your sources (including your own works), and quote or rephrase any duplicated text outside the methods section. Further consideration is dependent on these concerns being addressed.

“No funding”

6. Please include a separate caption for each figure in your manuscripta

Reviewers' comments:

Reviewer's Responses to Questions

**Comments to the Author**

1. Is the manuscript technically sound, and do the data support the conclusions?

Reviewer #1: Yes

Reviewer #2: Partly

2. Has the statistical analysis been performed appropriately and rigorously? 

Reviewer #1: I Don't Know

Reviewer #2: No

3. Have the authors made all data underlying the findings in their manuscript fully available?

Reviewer #1: Yes

Reviewer #2: Yes

4. Is the manuscript presented in an intelligible fashion and written in standard English?

Reviewer #1: Yes

Reviewer #2: Yes

5. Review Comments to the Author

Reviewer #1: The introduction in the abstract is long and needs to be shortened.

The type of study should be included in the method.

There is no information about existing studies with a similar objective. In this case, the first preference is to conduct studies in similar geographical areas, and the second preference is to conduct studies in other parts of the world where home birth is common.

What has been the shortcoming of the existing studies that the authors have investigated this relationship with the same goal.

Psychological factors such as feeling less fear at home, more comfort of the mother at home and perhaps cultural factors involved in choosing to give birth at home were not seen in this study. It is necessary to consider the limitations of this study.

The results and method are very well written.

It is necessary to write in the method on what basis the leveling is done and what variables each level (such as the individual level) includes? Is this stratification statistical or conceptual or both? It is necessary to address it in the introduction and in the method.

The results need to be reviewed by a statistician.

The flaw that exists in the discussion of the study is that the statistical index (odds ratio) obtained from the present study has not been compared with any other study.

It is necessary to review the study with conflicting results. In the following, the reason for the difference will be explained.

Reviewer #2: Title: The title is too long and seems to be carrying terms that are not relevant and can be included in the objectives e.g. if the authors remove the terms prevalence and determinants as well as the survey statement, then include these in the objectives and methodology, respectively.

Lines 140 and 142 are a repetition of "sex of household head".

Lines 140 and 145 are also a repetition of "wealth index".

Lines 186 and 194 have the spelling for Sub wrong and put as Sab.

The font size is not uniform- especially in the following lines- 276 and 360.

Line 305 statement is not consistent with the information in the rest of the paragraph. Maybe the authors meant ".......associated with hospital delivery" not "....home delivery" which they put in that statement.

Statistical analysis: I think there is too much statistics on ORs and this tends to to be very confusing and will limit readership to only those who are well conversant with detailed statistics. I propose the details on ORs and tables be trimmed and summarized to make it more clearer and understandable to majority of the audience.

6. PLOS authors have the option to publish the peer review history of their article (what does this mean?). If published, this will include your full peer review and any attached files.

Reviewer #1: **Yes: **leila Amiri

Reviewer #2: **Yes: **Dr Grace Danda

---

## [Author Response · Author response to Decision Letter 0]

6 Nov 2023

Please see "responses to reviewers" made by the authors.

---

## [Decision Letter · Decision Letter 1]

20 Nov 2023

Home delivery among women who had optimal ANC follow-up in Sub-Saharan Africa: A Multilevel Analysis

PONE-D-23-19452R1

Dear Dr. Alebachew,

We’re pleased to inform you that your manuscript has been judged scientifically suitable for publication and will be formally accepted for publication once it meets all outstanding technical requirements.

Kind regards,

Kahsu Gebrekidan

Academic Editor

PLOS ONE

Additional Editor Comments (optional):

Reviewers' comments:

Reviewer's Responses to Questions

**Comments to the Author**

1. If the authors have adequately addressed your comments raised in a previous round of review and you feel that this manuscript is now acceptable for publication, you may indicate that here to bypass the “Comments to the Author” section, enter your conflict of interest statement in the “Confidential to Editor” section, and submit your "Accept" recommendation.

Reviewer #1: (No Response)

Reviewer #2: All comments have been addressed

2. Is the manuscript technically sound, and do the data support the conclusions?

Reviewer #1: (No Response)

Reviewer #2: Yes

3. Has the statistical analysis been performed appropriately and rigorously? 

Reviewer #1: (No Response)

Reviewer #2: Yes

4. Have the authors made all data underlying the findings in their manuscript fully available?

Reviewer #1: (No Response)

Reviewer #2: Yes

5. Is the manuscript presented in an intelligible fashion and written in standard English?

Reviewer #1: (No Response)

Reviewer #2: Yes

6. Review Comments to the Author

Reviewer #1: (No Response)

Reviewer #2: The authors have clearly addressed all the areas that I had highlighted earlier. The statistical analysis query has also been explained and rationalized.

The article seems perfect and publishable.

7. PLOS authors have the option to publish the peer review history of their article (what does this mean?). If published, this will include your full peer review and any attached files.

Reviewer #1: **Yes: **leila AmiriFarahani

Reviewer #2: **Yes: **Dr Grace Danda

---

## [Editor Report · Acceptance letter]

23 Nov 2023

PONE-D-23-19452R1 

Home delivery among women who had optimal ANC follow-up in Sub-Saharan Africa: A Multilevel Analysis 

Dear Dr. Zegeye:

I'm pleased to inform you that your manuscript has been deemed suitable for publication in PLOS ONE. Congratulations! Your manuscript is now with our production department. 

Kind regards, 

on behalf of

Dr. Kahsu Gebrekidan 

Academic Editor

PLOS ONE